# Interactions between Microalgae and Bacteria in the Treatment of Wastewater from Milk Whey Processing

**Francesca Marazzi [1], Micol Bellucci [2], Tania Fantasia [1], Elena Ficara [2] and Valeria Mezzanotte [1,*]**

[1] Università Degli Studi di Milano—Bicocca, Department of Earth and Environmental Sciences (DISAT), P.zza Della Scienza 1, 20126 Milano, Italy; francesca.marazzi@unimib.it (F.M.); t.fantasia@hotmail.it (T.F.)

[2] Politecnico di Milano, Department of Civil and Environmental Engineering (DICA), P.zza L. da Vinci 32, 20133 Milano, Italy; micol.bellucci@polimi.it (M.B.); elena.ficara@polimi.it (E.F.)

\* Correspondence: valeria.mezzanotte@unimib.it; Tel.: +39-02-64482736

**Abstract:** Milk whey processing wastewaters (MWPWs) are characterized by high COD and organic nitrogen content; the concentrations of phosphorus are also relevant. A microalgal-based process was tested at lab scale in order to assess the feasibility of treating MWPW without any dilution or pre-treatment. Different microalgal strains and populations were tested. Based on the obtained results, *Scenedesmus acuminatus* (SA) and a mixed population (PM) chiefly made of *Chlorella, Scenedesmus,* and *Chlamydomonas* spp. were grown in duplicate for 70 days in Plexiglas column photobioreactors (PBRs), fed continuously (2.5 L culture volume, 7 days hydraulic retention time). Nutrient removal, microalgae growth, photosynthetic efficiency, and the composition of microalgal populations in the columns were monitored. At steady state, the microalgal growth was similar for SA and PM. The average removal efficiencies for the main pollutants were: 93% (SA), 94% (PM) for COD; 88% (SA) and 90% (PM) for total N; and 69% (SA) and 73% (PM) for total P. The residual pollution levels in the effluent from the PBRs were low enough to allow their discharge into surface waters; such good results were achieved thanks to the synergy between the microalgae and bacteria in the $CO_2$ and oxygen production/consumption and in the nitrogen mineralization.

**Keywords:** milk whey processing wastewater; chlorophyll fluorescence; photosynthetic efficiency; nutrient removal; microalgae–bacteria synergy

## 1. Introduction

The dairy industry includes a number of processes, starting from raw milk and generating different products to be used as food, feed, or ingredients in the food industry (pasteurized and sour milk; yoghurt; hard, soft, and cottage cheese; cream and butter products; ice cream; milk and whey powders; lactose; condensed milk; as well as various types of desserts) [1]. In Europe, more than 170.1 million tons of raw milk are produced and 158.6 million tons of milk are delivered to dairies to be processed into fresh products (drinking milk, yoghurts, cream, fermented milks) and manufactured products (cheese, milk powder, butter, whey, etc.) [2].

Of course, the water demand of the dairy industry is high, as water is needed in all of the steps of milk processing, and large volumes are consumed in washing operations. As a consequence, the amount of wastewater to be treated and disposed is huge, and is strictly dependent on the factory size, the applied technology, and the overall management of the processes [3,4]. According to Slavov [1], wastewater production ranges between 0.5 to 37 $m^3/m^3$ of processed milk, with an average of 2.5. More recently, narrower ranges have been indicated: Ahmad et al. [5] reported water consumptions between 6 and 10 $m^3/m^3$ of processed milk.

On such a basis, dairy processing is considered as one of the main wastewater sources, especially in Europe [3,4,6–8].

The composition of dairy wastewaters is also quite variable, but some common features can be identified, such as the high concentrations of organic matter (especially made of lactose, oil and grease, proteins), nitrogen and suspended solids [9,10], and the presence of various trace soluble organics. Various residues of cleaning products, including alkaline and acidic chemicals, are also often present. So, COD is high (80–95 g/L), but BOD is high too (40–48 g/L) due to the presence of biodegradable substances [11]. The concentrations of suspended solids, TKN, and total phosphorus range between 0.1 and 22.0 g/L, 0.01 and 1.7 g/L, and 0.006 and 0.5 g/L, respectively [11]. According to Ahmad et al. [5], pH also varies within a very wide range (4.7–11). Dairy effluents often include milk or milk products lost during processing (skimmed milk, spoiled milk, spilled milk, and curd pieces), by-products of processing operations (whey permeates, whey, and milk), starter cultures used in manufacturing of fermented products, reagents used in CIP (cleaning in-place) procedures, contaminants used for washing trucks, cans, equipment tanks, bottles and floors, and different additives used in the manufacturing process [1].

Whey also is largely lost and contributes to the organic and protein load in wastewater [12]. It is considered as the most important pollutant in dairy wastewaters due to its high organic load. Whey processing wastewaters have the same composition as whey, but at lower concentrations due to the high flow and, consequently, to the high dilution. Besides the usual components of dairy effluents, whey processing wastewater may contain minor components such as citric and lactic acids (0.02–0.05%), non-proteinic nitrogen compounds (urea and uric acid), and vitamins (B group) [13].

Different kinds of processes and different process sequences are used to treat dairy effluents, especially in order to remove nitrogen, according to the site-specific production. Among biological processes, the aerobic ones are often adopted, but their efficiency is limited by the rapid acidification they cause, fostered by the low buffer capacity of dairy wastewater, and by the growth of filamentous bacteria, fostered by the presence of high levels of lactose. Due to their flexibility, sequencing batch reactors (SBRs) are often chosen [1].

Anaerobic processes seem a better and more cost-effective choice due to the high organic and fat concentration in dairy wastewaters [11], and UASB (Upflow Anaerobic Sludge Blanket) reactors are often reported as the preferable option [5,14].

Microalgae/bacteria consortia have been recently suggested as effective in the treatment of different kinds of urban [15–18], as well as agro-industrial [19], wastewater. According to some authors, microalgae-based processes are not suitable to treat dairy wastewater as such, due to the high polluting load they carry [20,21]. Indeed, most of the experiments were performed on diluted or pre-treated wastewater [22–24]. The aim of the present research was to test the possibility of using spontaneous consortia made of microalgae and bacteria for treating wastewater from a milk whey processing factory without any pre-treatment or dilution.

## 2. Materials and Methods

### 2.1. Wastewater

The tested wastewater (milk whey processing wastewater, MWPW) was collected from a factory processing milk whey to recover lactose and casein to be used as animal feed. The process cycle includes:

- Reverse osmosis,
- Ultrafiltration,
- Concentration by MVR (mechanical vapor recompression) evaporation.

Cleaning operations involve the use of acids, bases, and disinfectants and produce about 1000 m³/day of wastewater. These are stored in an equalization tank (843 m³) and undergo aerobic

biological treatment based in a membrane bioreactor. The amount of wastewater needed for the whole trial was collected at the beginning of the experimentation and stored in a cold chamber (5–6 °C). The characterization of the raw wastewater used for the experimental trial is reported in Table 1.

**Table 1.** Characterization of the raw wastewater treated by microalgae–bacteria consortium (*n* = 3).

| Parameter | Value | Parameter | Value |
|---|---|---|---|
| pH | 7.8 ± 0.16 | Ca (mg/L) | 177 ± 64 |
| Electric conductivity (µS/cm) | 3312 ± 227 | Si (mg/L) | 4 ± 0.4 |
| Turbidity (FAU) | 87 ± 19 | Fe (mg/L) | 2.3 ± 0.4 |
| Total suspended solids (TSS; mg/L) | 180 ± 40 | Na (mg/L) | 595 ± 66 |
| Volatile suspended solids (VSS; mg/L) | 130 ± 20 | Zn (mg/L) | 0.02 ± 0.01 |
| Total N (mg/L) | 52 ± 7 | Mn (mg/L) | 0.014 ± 0.02 |
| $NH_4$–N (mg/L) | 31 ± 6 | Mo (µg/L) | <0.1 |
| $NO_3$–N/mg/L) | 0.3 ± 0.04 | Al (µg/L) | <0.1 |
| $NO_2$–N (mg/L) | <0.3 | Cr (µg/L) | 1.4 ± 0.7 |
| Total P (mg/L) | 17 ± 1.5 | Ni (µg/L) | 7.8 ± 8.5 |
| $PO_4$–P (mg/L) | 23.0 ± 1.3 | Pb (µg/L) | 29 ± 4 |
| COD (mg/L) | 982 ± 253 | Cu (µg/L) | 21 ± 16 |
| K (mg/L) | 87 ± 9 | Cd (µg/L) | 0.3 ± 0.4 |
| Mg (mg/L) | 15 ± 2 | $Cl^-$ (mg/L) | 591 ± 260 |
| | | $SO_4^-$ (mg/L) | 33 ± 11 |

The wastewater was rich in nutrients needed to support microalgal growth, and the N/P molar ratio was 4, largely below the Redfield ratio [25,26]. Turbidity and solid contents were low, suggesting adequate optical properties. MWPW did not undergo any pre-treatment, and thus contained a rich bacterial population.

## 2.2. Analytical Determinations

Ammoniacal, nitrite, and nitrate nitrogen ($NH_4$, $NO_3$, and $NO_2$), phosphate (P–$PO_4$), and soluble COD were measured using spectrophotometric test kits (Hach-Lange, Düsseldorf, Germany, LCK 303, LCK 340, LCK 342, LCK 348, and LCK1414, respectively), on 0.45 µm filtered samples. For the batch tests, pH and conductivity were measured by a portable probe (XS PC 510 Eutech Instruments, Stevensville, MI, USA). In the continuous test, pH was measured online and recorded. Microalgal growth and density were determined by direct and indirect methods. Direct counts of microalgal cells were carried out using a hemocytometer (Marienfeld, Lauda-Königshofen, Germany) and a microscope (B 350, Optika, Ponteranica, Italy). A 1 mL sample of microalgal suspension was collected from each photobioreactor and diluted 1:10; then, 0.1 mL of the sample was injected into the hemocytometer chamber. *Scenedesmus*, *Chlorella*, and *Chlamydomonas* cells were distinguished according to their morphological characteristics, and then counted. The final estimated cell number was obtained from the mean of 9 square (1 mm$^2$) readings. Indirect assessments were based on the determinations of total and volatile suspended solids (TSS and VSS), according to standard methods [27], and of optical density (OD), measured by a DR 3900 Hach Lange (Germany) spectrophotometer at a wavelength of 680 nm using a 1 cm path length cuvette.

The elemental analysis on dried samples was performed by a Perkin Elmer CHNS/O analyzer 2400 series II. Phosphorus was determined after acid digestion (with $HNO_3$ and $H_2O_2$) of the dry biomass in a microwave digester (ETHOS 1600, Milestone, Italy) according to the Green Algae Procedure (DG-EN-25).

## 2.3. Photosynthetic Efficiency

Samples of the algal suspensions were collected two or three times a week (69 samples on the whole) and diluted to 0.1 optical density (at 680 nm). The suspension was then kept in the dark for

20 min before starting the PAM measurement, which was performed by Phyto-PAM II (Heinz Walz GmbH, Effeltrich, Germany).

$F_0$ (minimal fluorescence yield of dark-adapted sample with all PS II centers open) was determined after an acclimation period of 45 s at very low intensity light (PAM set intensity = 2, corresponding to PAR = 1 $\mu$mol m$^{-2}$ s$^{-1}$). $F_m$ (maximal fluorescence yield of dark-adapted sample with all PS II centers closed) was measured after a saturation light pulse (PAM set intensity = 10, corresponding to PAR $\approx$ 26 $\mu$mol m$^{-2}$ s$^{-1}$ width 500 ms).

Photosynthetic efficiency is an adimensional value, determined by variable ($F_v$) and maximum ($F_m$) fluorescence according to Kitajima and Butler [28] as follows:

$$F_v/F_m = (F_m - F_0)/F_m \tag{1}$$

The Phyto-Pam-II deconvolution function was used to distinguish fluorescence signals emitted by four different groups of photosynthetic organisms: cyanobacteria, green algae, diatoms/dinoflagellates, and phycoerythrin-containing organisms. Therefore, the deconvoluted $F_0$ values were used as a proxy of different microalgae proportions in the suspensions.

### 2.4. Preliminary Screening Test

Small-scale tests were carried out using 50 mL Erlenmeyer flasks, put on an orbital shaker, artificially illuminated at 20 $\mu$mol m$^{-2}$ s$^{-1}$ (12 h/12 h light/dark cycle), inoculated with:

- *Chlorella* sp.
- *Scenedesmus acuminatus*
- *Scenedesmus obliquus*
- *Arthrospira platensis* (Spirulina)
- A mixed microalgal population made of *Chlorella, Chlamydomonas*, and *Scenedesmus* spp.

The test was carried out for 27 days. During the first 10 days, microalgae were grown in Bold's Basal Medium (BBM) [29]. Then, a gradual acclimation was searched by replacing 40% of the BBM medium with MWPW after 10 days, and repeating this operation three times at 6 day intervals. Microalgal growth was monitored by cell counts and absorbance at 680 nm for 17 days after the first addition of MWPW.

### 2.5. Continuous Test

The continuous test was carried out for 70 days in a fully equipped apparatus for microalgae culturing (IDEA Bioprocess Technology Srls, Dalmine, BG, Italy) (Figure 1). This apparatus was made of four parallel Plexiglas column photobioreactors (PBRs) having a diameter of 110 mm, a height of 40 cm, and a working volume of 2.5 L. The system was equipped with artificial white lights (about 100 $\mu$mol m$^{-2}$ s$^{-1}$) operated according to 12 h/12 h light/dark cycles. A feeding pump was allowed to operate continuously with a hydraulic retention time (HRT) of 7 days. Temperature remained at 26.6 $\pm$ 2.6 °C. The photobioreactors were mixed by magnetic stirrers and pH was controlled by pH-driven bubbling of $CO_2$ (set point pH = 7.5).

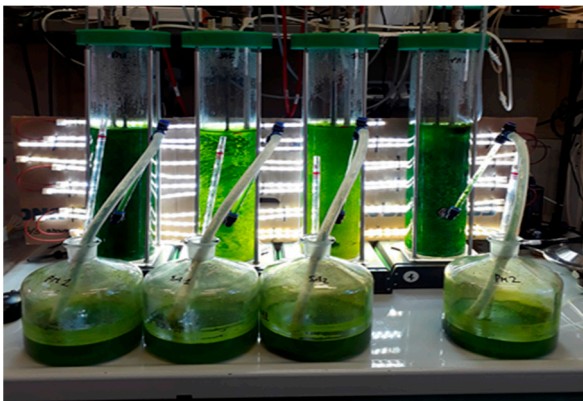

**Figure 1.** Column photobioreactors (PBRs) used for lab-scale microalgal culturing.

The PBRs were filled with MWPW and inoculated with suspensions of *Scenedesmus acuminatus* (SA1 and SA2) and of a mixed population (PM1 and PM2), selected during the batch tests, with an initial cell count of $1.2 \times 10^6$ cells/mL in each reactor.

*2.6. Statistical Analyses*

The *t*-test for paired data was performed to detect differences between the algae cultures (SA vs. PM) for microalgae cell counts and nutrient removal rates (mg/L/d). *p*-values <0.05 were deemed to be statistically significant.

## 3. Results

*3.1. Selection of Microalgal Strains and Populations*

In terms of microalgal growth (based on cell counts and absorbance at 680 nm), the best results were obtained with *Scenedesmus acuminatus* (SA), and with the mixed population (PM). The final OD680 values were 1.14 and 1.88, respectively, while the OD680 of all the other cultures was below 0.5. In both cases, the maximum OD680 was measured 8 days after the input of MWPW, and thereafter, values remained nearly constant. Therefore, SA and PM were selected as inocula to be tested in the following continuous test.

*3.2. Continuous Test*

Microalgal growth showed a certain variability, but a substantial stability can be observed from day 21, as shown in Figures 2 and 3 reporting the values of optical density (at 680 nm) and the cell counts, respectively. The microscope cell counts remained between $10^6$ and $10^7$ cells/mL throughout the test.

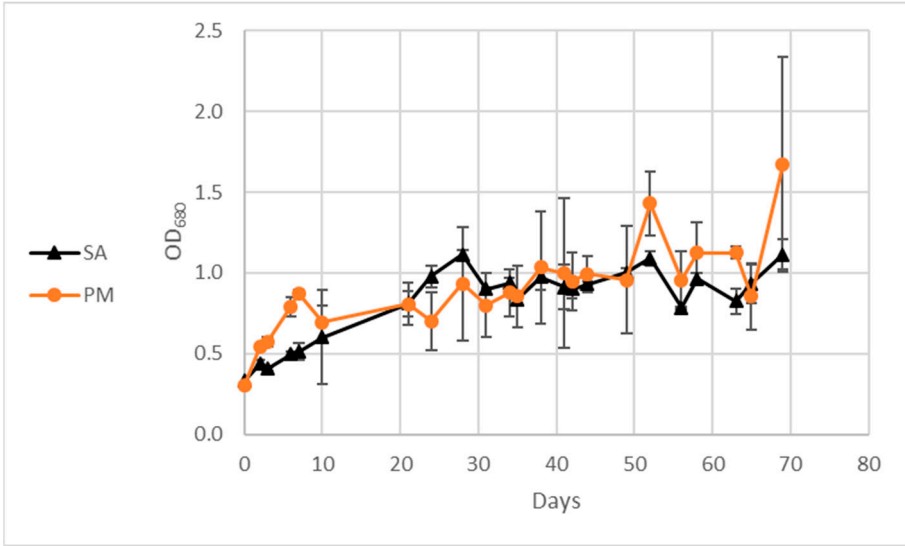

**Figure 2.** Values of optical density at 680 nm in the PBRs during the test. Average and standard deviations for the two replicates of *Scenedesmus acuminatus* (SA) and mixed population (PM).

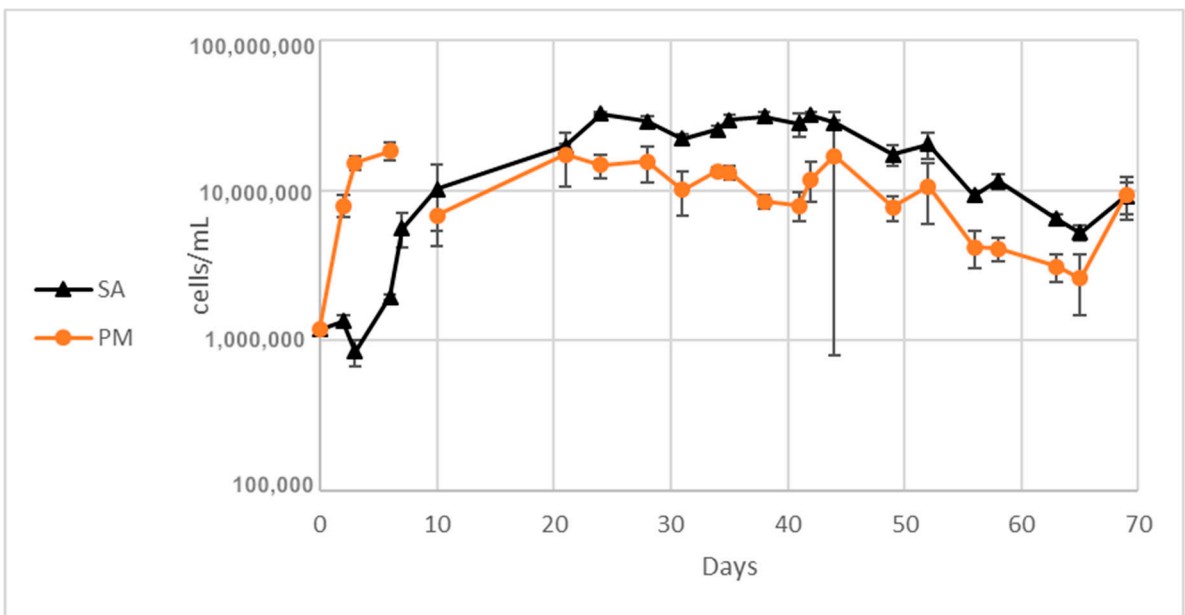

**Figure 3.** Cell counts in the PBRs during the test. Average and standard deviations for the two replicates of *Scenedesmus acuminatus* (SA) and mixed population (PM).

As shown in Figure 4, the concentration of total suspended solids, including both microalgae and bacteria biomass, also stabilized after approximately 21 days. The replicability of this dataset was suboptimal and negatively affected by the presence of aggregates.

As shown in Figure 5, a contamination by cyanobacteria took place in all the photobioreactors. However, in the case of SA, cyanobacteria never exceeded 10% of the total population, while in PM photobioreactors, cyanobacteria grew steadily from day 30 until becoming the dominant group of organisms from day 60. The filamentous nature of cyanobacteria may have favored aggregate formation. Indeed, solid/liquid separation of the algal/bacteria biomass was very efficient and fast, opposite to the typical poor settleability of algal biomass, involving the need for expensive harvesting processes [30]. As regards nutrient removal, cyanobacteria perform similarly to microalgae, but their cells often have more interesting properties in the perspective of biomass valorization.

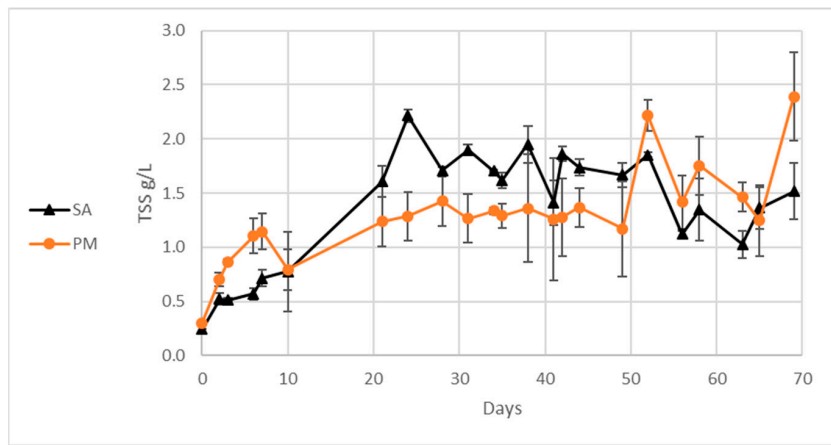

**Figure 4.** Concentrations of total suspended solids (TSS, g/L) in the PBRs during the test. Average and standard deviations for the two replicates of *Scenedesmus acuminatus* (SA) and mixed population (PM).

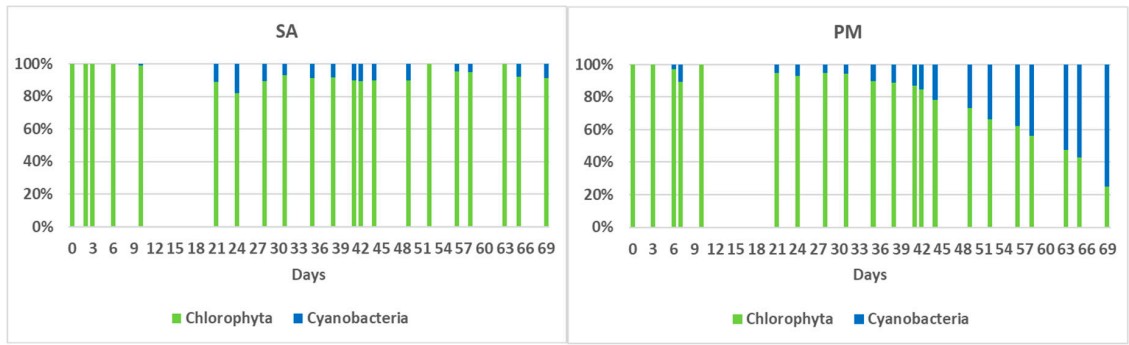

**Figure 5.** Deconvolution of green algae and cyanobacteria in SA and PM photobioreactors during the continuous test. Average and standard deviations for the two replicates of *Scenedesmus acuminatus* (SA) and mixed population (PM).

The trial conditions appeared suitable, as shown by PhytoPAM analyses (Figure 6). Indeed, Fv/Fm ratio ranged between 0.5 and 0.7, apart from the very beginning of the test, where a transient drop of photosynthetic efficiency in SA photobioreactors occurred, suggesting that some acclimation time was probably needed (Figure 6). Nevertheless, all values are considered optimal according to Ranglová et al. [31].

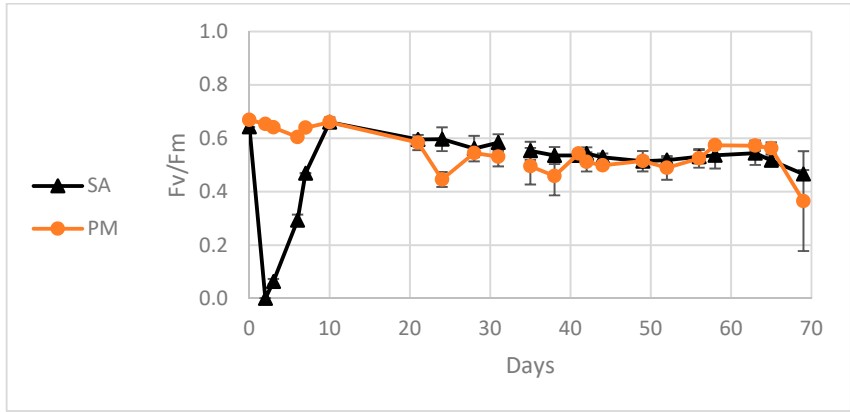

**Figure 6.** Photosynthetic efficiency, expressed as Fv/Fm ratio, during the continuous test. Average and standard deviations for the two replicates of *Scenedesmus acuminatus* (SA) and mixed population (PM).

As shown in Figure 7, a relevant removal of COD and N occurred during the initial days of continuous operation. Later, concentrations remained steadily low all over the test. At steady state, the average removal efficiency was 93% in SA PBRs and 94% in PM PBRs for COD, and 88% and 90%, respectively, for total N. The removal efficiency of phosphorus was more variable, with average values of 69% and 73% in SA and PM PBRs, respectively.

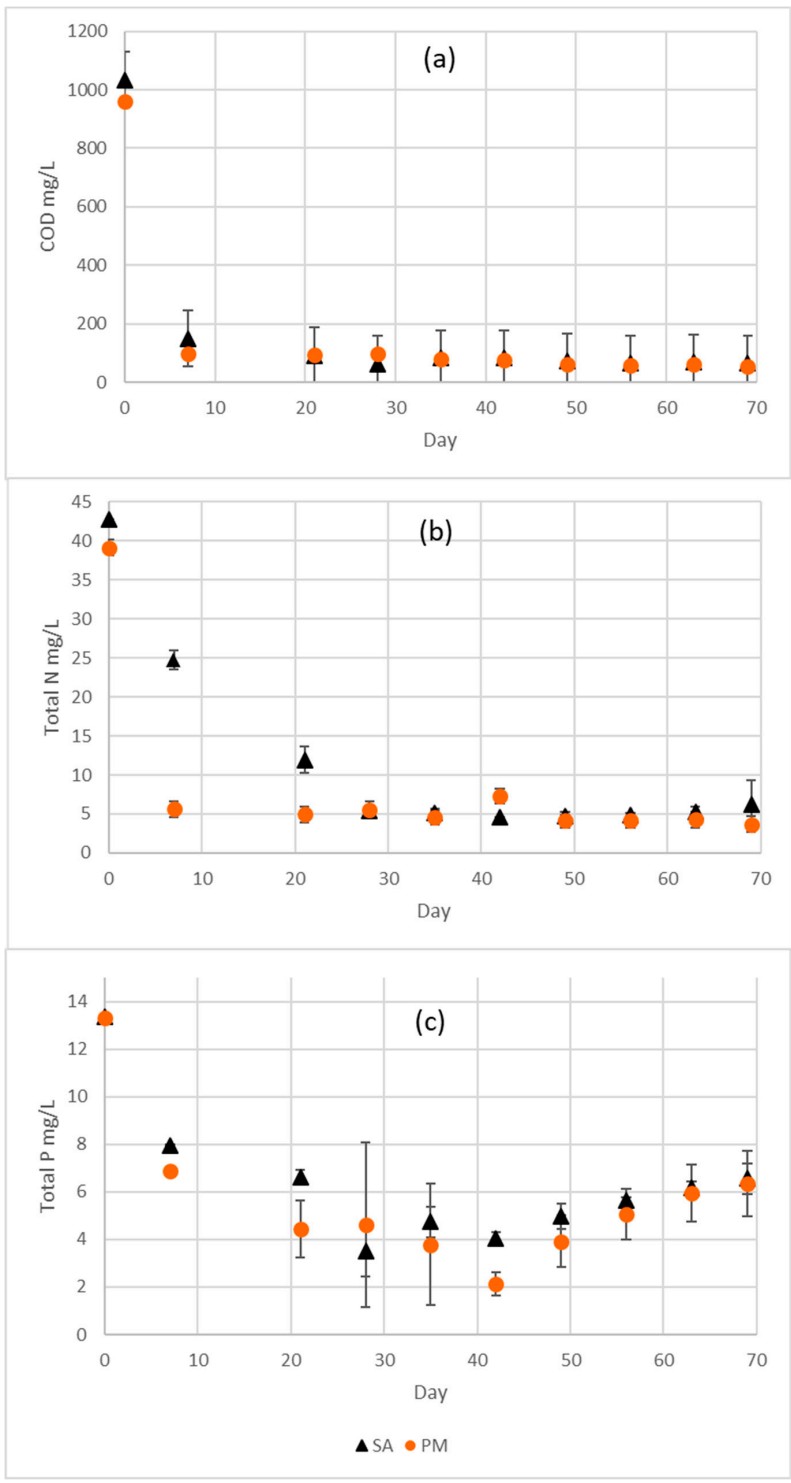

**Figure 7.** COD (**a**), total N (**b**), and total P (**c**) concentrations in the PBRs during the continuous test. Average and standard deviations for the two replicates of *Scenedesmus acuminatus* (SA) and mixed population (PM).

The average removal of $NH_4^+$–N was 88% and 99% in SA and PM photobioreactors, respectively. The *t*-test showed that the removal efficiency of COD, total N, total P, and $NH_4^+$–N was not significantly different between SA and PM.

In spite of some differences in the composition of microalgae populations, the biomass composition was very similar in the four PBRs. The elemental analysis indicated the following concentrations: C = 35 ± 2%, H = 5 ± 0.4%, N = 6 ± 1.0%, P 11 ± 1‰ in SA PBRs; C = 37 ± 1%, H = 6 ± 0.3%, N = 6 ± 0.3%, and P = 11 ± 1‰ in PM PBRs.

## 4. Discussion

The residual concentrations of the main pollutants (COD, total N, $NH_4^+$–N, P) observed in the effluent of the PBRs were low enough to allow their discharge into surface water according to the Italian regulations, as shown in Table 2. The survey was not extended to micropollutants, whose presence is not likely in this kind of process wastewater, nor to heavy metals, whose concentration in the MWPW was already very low. Based on the high bacterial counts in the PBRs, any detrimental effect on biological populations due to the disinfectants used in the industrial process and possibly present in the MWPW could be excluded. Furthermore, the absence of foam during the whole continuous test, in spite of continuous stirring, suggests the absence of significant concentrations of surfactants.

**Table 2.** Comparison between the ranges of values measured in the effluents from the photobioreactors (PBRs) and the limits set by Italian law (D.Lgs.152/2006) for the discharge into surface water.

| Parameter | Value in Effuents from PBRs | Italian Limits for Discharge in Surface Waters |
|---|---|---|
| pH | 7.5 | 5.5–9.5 |
| COD (mg/L) | 43–102 | 160 |
| $NH_4$–N (mg/L) | 0.14–3.0 | 15 |
| $NO_3$–N (mg/L) | <0.3 | 20 |
| $NO_2$–N (mg/L) | <0.6 | 0.6 |
| Total P (mg/L) | 2.0–7.0 | 10 |

The pH-controlled supply of $CO_2$ was almost negligible (0.2 mg $CO_2$/L/day on average): The increase of pH due to photosynthesis was possibly buffered by the production of $CO_2$ from bacterial activity and that could be enough to keep the pH constant.

These results confirm that the microalgae–bacteria consortium grew regularly in the tested MWPW and removed pollutants to such an extent as to obtain a good quality effluent. In the experimental conditions, microalgal mixotrophic metabolism was unlikely to have played a significant role, due to the light exposure and to the relevant presence of bacteria in the fed MWPW.

At the steady state, starting from day 21, the average removal rates of total nitrogen were 6.4 and 6.5 mg $L^{-1}$ $d^{-1}$ in SA and PM PBRs. For $NH_4$–N, whose concentrations in the fed MWPW were lower, the removal rates were 4.1 and 4.2 mg $L^{-1}$ $d^{-1}$, respectively. However, considering the effective and stable N removal performances, higher nitrogen removal rates can likely be achieved by reducing the hydraulic retention time. This good performance was likely achieved by the synergy between microalgae and bacteria. In fact, phosphorus was not limiting and even exceeded the microalgal demand, and ammoniacal nitrogen was 60% of total nitrogen. Under these non-limiting nutrient conditions, bacterial activity was intense and resulted in a substantial biodegradation of organic matter (including organic nitrogen) and to the release of $CO_2$, which could be used by microalgae.

In order to better understand the interaction between algae and bacteria, a steady state calculation of the expected heterotrophic bacteria concentration in the suspension was carried out according to the

conventional theory on biological oxidation of degradable organics [32,33]. Specifically, the steady state equation applied to quantify the concentration of heterotrophic biomass ($X_B$) is

$$[X_B] = \frac{Y \cdot ([S]_{IN} - [S]_{OUT})}{1 + b' \cdot HRT} \text{ with } b' = b_{20} \cdot \theta^{T-20} \cdot [1 - (1 - f) \cdot Y] \qquad (2)$$

where Y is the growth yield for heterotrophic bacteria (0.45 g TSS/g COD); $[S]_{IN}$ is the soluble COD in the fed MWPW (982 mg COD/L); $[S]_{OUT}$ is the soluble COD in the effluent from the PBRs (69 mg COD/L for SA and 60 mg COD/L for PM); b' is the net decay coefficient (i.e., the decay constant corrected for the cryptic growth); b is the decay constant (0.12 $d^{-1}$ at 20 °C); $\theta$ is the temperature correction factor for the decay constant (1.07); f is the fraction of cell debris released from biomass decay (0.12 g/g). According to Equation (2), the expected heterotrophic biomass concentration is 282 mg TSS/L for SA and 285 mg TSS/L for PM.

From bacteria decay ($X_P$), inert cell debris are also produced and quantified as follows:

$$[X_P] = f \cdot b_{20} \cdot \theta^{T-20} \cdot HRT \cdot [X_B] \qquad (3)$$

According to Equation (3), a cell debris concentration of 44.9 mg TSS/L for SA and 45.4 mg TSS/L for PM is quantified.

The expected oxygen request (OR) for COD oxidation was computed considering both exogenous and endogenous terms as follows:

$$OR = \frac{([S]_{IN} - [S]_{OUT}) \cdot (1 - Y)}{HRT} + (1 - Y) \cdot b_{20} \cdot \theta^{T-20} \cdot [X_B] \cdot (1 - f) \qquad (4)$$

According to Equation (4), the oxygen request was 46 mg $O_2$ $L^{-1}$ $d^{-1}$ for both algae populations. This value was compared with the oxygen production by algae (OP) that was estimated as follows:

$$OP = rTSS\_a \cdot \alpha_{O2} \qquad (5)$$

where $\alpha_{O2}$ is the specific oxygen production per unit of algal biomass produced (1.57 g $O_2 \cdot g^{-1}$ TSS, [20]); rTSS_a is the rate of algae biomass production which is quantified as the difference between the overall rate of biomass production rTSS and the rate of production of heterotrophic bacteria and cell debris, rTSS_b:

$$rTSS\_a = rTSS - rTSS\_b \qquad (6)$$

$$rTSS = \frac{[TSS]_{IN} - [TSS]_{OUT}}{HRT} \qquad (7)$$

$$rTSS\_b = \frac{[X_B] + [X_P]}{HRT} \qquad (8)$$

According to Equations (6)–(8), an algal biomass production rate, rTSS_a, of 56 and 70 mg TSS $L^{-1}$ $d^{-1}$ was obtained for SA and PM, respectively. By substituting these values in Equation (5), an oxygen production of 84 and 105 mg $O_2$ $L^{-1}$ $d^{-1}$ was obtained for SA and PM, respectively. By comparing the computed values for OP and OR, it is clear that photo-oxygenation fully compensates for the oxygen request for COD oxidation.

The heterotrophic $CO_2$ production rate, CP, is computed from the OR as follows:

$$CP = OP \cdot \alpha_{CO2\_O2} \qquad (9)$$

where $\alpha_{CO2\_O2}$ is the mass ratio between $CO_2$ and oxygen (1.375 = 44/32): Assuming that MWPW is mostly made of carbohydrates, the molar $CO_2$ production can be considered as equal to the molar oxygen request for COD oxidation. According to Equation (9), a $CO_2$ production rate of 62 mg $CO_2$ $L^{-1}$ $d^{-1}$ was obtained for both SA and PM PBRs. As for the algal $CO_2$ request, a stoichiometric $CO_2$

request of 1.88 g $CO_2$/g TSS was assumed, leading to 106 and 132 mg $CO_2$ $L^{-1}$ $d^{-1}$ for SA and PM, respectively. According to these results, $CO_2$ production from COD oxidation by bacteria could cover up to 58% of the algae request, thus suggesting that the remaining inorganic carbon was taken from the small amount of $CO_2$ supplied by the system, from the incoming alkalinity, from the atmosphere, and, possibly, from other processes occurring inside the PBRs as, for instance, urea hydrolysis.

The chief aspects of the mass balance concerning the interactions between microalgae and bacteria are summarized in Figure 8.

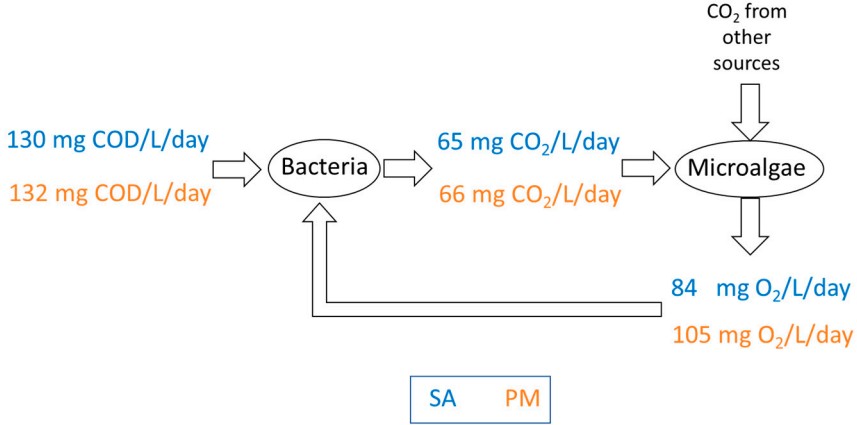

**Figure 8.** Mass balance of carbon and oxygen in the studied system.

Organic nitrogen was converted to ammoniacal nitrogen, which was taken up by microalgae and bacteria; no nitrification occurred. The absence of nitrification was likely due to the dominance of heterotrophic bacteria, using the large amount of biodegradable organic matter as a substrate, and of microalgae, more competitive for $CO_2$ uptake.

## 5. Conclusions

Experimental results on the treatability of milk whey processing wastewaters were promising. Specifically, the following conclusions can be drawn:

- *Scenedesmus acuminatus* (SA) and a mixed population of *Chlorella, Scenedesmus,* and *Chlamydomonas* spp. (PM) could grow in batch tests on MWPW, unlike other tested pure cultures (*Chlorella* sp., *Scenedesmus obliquus, Arthrospira platensis*).
- MWPW was effectively treated in a 70-day continuous test using both SA and PM and effluent pollution level complied with limits set by the Italian law for discharge into surface water.
- Continuously operated PBRs were contaminated by cyanobacteria, whose filamentous nature may have favored biomass aggregation, thus promoting efficient harvesting by gravity settling.
- According to steady state calculations, photo-oxygenation fully compensated for the oxygen request for COD oxidation; on the contrary, $CO_2$ production from COD oxidation by bacteria could cover up to 58% of the algae request, suggesting that the remaining inorganic carbon was taken from other sources (e.g., bubbled $CO_2$, incoming alkalinity, gas exchange with the atmosphere).

**Author Contributions:** Conceptualization, V.M. and E.F.; methodology, F.M. and M.B.; formal analysis, T.F.; investigation, M.B.; data curation, F.M.; writing—Original draft preparation, V.M.; writing—Review and editing, F.M., M.B., T.F., and E.F.; supervision, V.M. All authors have read and agreed to the published version of the manuscript.

**Funding:** This research was funded by FONDAZIONE CARIPLO, grant title "Il Polo delle Microalghe".

**Acknowledgments:** The authors thank SERUM Italia SpA for providing the wastewater and CREA for preliminary data on microbiological counts.

**Conflicts of Interest:** The authors declare no conflicts of interest. The funders had no role in the design of the study; in the collection, analyses, or interpretation of data; in the writing of the manuscript; or in the decision to publish the results.

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
