# Peer review of "Interactions between Microalgae and Bacteria in the Treatment of Wastewater from Milk Whey Processing"

_water, doi:10.3390/w12010297_

Round 1

Reviewer 1 Report

The manuscript entitled "Interactions microalgae-bacteria in the treatment of wastewater from milk whey processing" deals with the use of specific microalgal populations with the purpose of analysing their behaviour on real dairy wastewaters and studying their capability to remove nutrients and COD from such wastewaters. The topic is interesting and being widely studied especially in Mediterranean countries. Despite having some grammatical mistakes that should be urgently revised, the current manuscript is well written and could catch the attention of potential readers. For all these reasons together with the below-mentioned specific comments, I strongly recommend a minor revision of this manuscript.

Some specific comments are:

1) English should be polished.

2) Authors claim that the microalgal growth presented a steady trend since day 21 until the end of the experiment, but it can be observed in both figures 2 and 3 that the average of the results and its variability remarkably increased in terms of total suspended solids since day 50. Could authors explain this contradiction?

3) Some huge differences in the results concerning standard deviation are observed in the results showed in figure 4. Could explain that?

Author Response

1) English should be polished. The manuscript has been fully revised with the help of an English translator

2) Authors claim that the microalgal growth presented a steady trend since day 21 until the end of the experiment, but it can be observed in both figures 2 and 3 that the average of the results and its variability remarkably increased in terms of total suspended solids since day 50. Could authors explain this contradiction? The variability after day 50 is chiefly related to an increase of microalgal density in PM and a decrease in SA PBRs. For PM data measured at days 52 and 69 are particularly high. Still, a high variability in all the values representing algal growth and density is normal and expected, as the analyses concern a non homogeneous suspension and, thus, it is very hard to collect perfectly representative samples.

3) Some huge differences in the results concerning standard deviation are observed in the results showed in figure 4. Could explain that? The only datum having a very high standard deviation has been produced at day 44 and we are really not able to explain that, which can be considered as an outlier. We have decided to leave it there because, in fact, even if the difference seems huge, the variation is hardly over one order of magnitude

Reviewer 2 Report

Comments to the Author and Editors

Water-462047

Manuscript title:

Interactions microalgae-bacteria in the treatment of wastewater from milk whey processing

Authors:

Francesca Marazzi, Micol Bellucci, Tania Fantasia, Elena Ficara, Valeria Mezzanotte

The aim of this paper is the analysis of the possibility of using spontaneous consortia made of microalgae and bacteria for treating wastewater from a milk whey processing factory without any pre-treatment or dillution.

The paper is interesting but requires minor changes.

Some specific comments: 

Specify clearly the purpose of the work in the manuscript and in abstract.

Please explain under the figures 2-7 what is this SA and PM

Page 3 – Table 1

Please give the concentration of Total Suspended Solids and Volatile Suspended Solids in mg/L instead g/L

Page 4 – Line 121

Please to give how many samples of treated wastewater under the research period were taken.

Page 4 - Line 147

Please explain why the test was carried out for only 27 days. On the figures 2-7 we can see that the results are from 70 days of research.

Page 8

Please divide chapter No 4. Discussion and conclusions into two chapters: 4. Discussion and 5. Conclusions

I would recommend accepting this manuscript for publishing after minor corrections. 

Author Response

Specify clearly the purpose of the work in the manuscript and in abstract. Sentences specifying the aim of the work have been included both n the Abstract (lines 13-15) and in the Introduction (lines 78-80).

Please explain under the figures 2-7 what is this SA and PM. It has been specified in the captions

Page 3 – Table 1 Please give the concentration of Total Suspended Solids and Volatile Suspended Solids in mg/L instead g/L The unit and corresponding values have been modified accordingly

Page 4 – Line 121 Please to give how many samples of treated wastewater under the research period were taken. It has been specified

Page 4 - Line 147 Please explain why the test was carried out for only 27 days. On the figures 2-7 we can see that the results are from 70 days of research. Batch tests for selecting the most suitable populations were carried out for 27 days, while continuous test lasted 70 days

Page 8

Please divide chapter No 4. Discussion and conclusions into two chapters: 4. Discussion and 5. Conclusions The chapter has been divided in two separate chapters for Discussion and Conclusions